# Mapping of morpho-electric features to molecular identity of cortical inhibitory neurons

**Yann Roussel**⦿\*, **Csaba Verasztó**⦿, **Dimitri Rodarie**⦿, **Tanguy Damart,
Michael Reimann**⦿, **Srikanth Ramaswamy**⦿, **Henry Markram, Daniel Keller**

Blue Brain Project, École polytechnique fédérale de Lausanne (EPFL), Geneva, Switzerland

\* yann.roussel@epfl.ch

**Data Availability Statement:** Code is under open sourcing process and is publicly available at https://github.com/BlueBrain/me-features-to-mo-ID-mapping. Downloading of Allen Institute for

## Abstract

Knowledge of the cell-type-specific composition of the brain is useful in order to understand the role of each cell type as part of the network. Here, we estimated the composition of the whole cortex in terms of well characterized morphological and electrophysiological inhibitory neuron types (me-types). We derived probabilistic me-type densities from an existing atlas of molecularly defined cell-type densities in the mouse cortex. We used a well-established me-type classification from rat somatosensory cortex to populate the cortex. These me-types were well characterized morphologically and electrophysiologically but they lacked molecular marker identity labels. To extrapolate this missing information, we employed an additional dataset from the Allen Institute for Brain Science containing molecular identity as well as morphological and electrophysiological data for mouse cortical neurons. We first built a latent space based on a number of comparable morphological and electrical features common to both data sources. We then identified 19 morpho-electrical clusters that merged neurons from both datasets while being molecularly homogeneous. The resulting clusters best mirror the molecular identity classification solely using available morpho-electrical features. Finally, we stochastically assigned a molecular identity to a me-type neuron based on the latent space cluster it was assigned to. The resulting mapping was used to derive inhibitory me-types densities in the cortex.

## Author summary

The computational abilities of the brain arise from its organization principles at the cellular level. One of these principles is the neuronal type composition over different regions. Since computational functions of neurons are best described by their morphological and electrophysiological properties, it is logical to use morpho-electrically defined cell types to describe brain composition. However, characterizing morpho-electrical properties of cells involve low-throughput techniques not very well suited to scan the whole brain. Thanks to recent progress on transcriptomic and immuno-staining techniques we are now able to get a more accurate snapshot of the mouse brain composition for molecularly defined cell types. How to link molecularly defined cell types with morpho-electrical cell types remains

Brain Science data was not incorporated in the code and is left at users discretion due to license issues. We recommend to use allensdk package (https://github.com/AllenInstitute/AllenSDK) to obtain the Allen Institute for Brain Science data. More details are provided in the README.md file of our public repository.

**Funding:** This study was supported by funding to the Blue Brain Project, a research center of the École polytechnique fédérale de Lausanne (EPFL), from the Swiss government's ETH Board of the Swiss Federal Institutes of Technology. The funders had no role in study design, data collection and analysis, decision to publish, or preparation of the manuscript.

**Competing interests:** The authors have declared that no competing interests exist.

an open question. Several studies have explored this problem providing valuable three-modal datasets combining electrical, morphological and molecular properties of cortical neurons. The long-term goal of the Blue Brain Project (BBP) is to accurately model the mouse's whole brain, which requires detailed biophysical models of neurons. Instead of going through the time-consuming process of producing detailed models from the three-modal datasets, we explored a time-saving method. We mapped the already available detailed morpho-electrical models from the BBP rat dataset to cells from a three-modal mouse dataset. We thus assigned a molecular identity to the neuron models allowing us to populate the whole mouse cortex with detailed neuron models.

This is a *PLOS Computational Biology* Methods paper.

## Introduction

The ability of the brain to produce complex computational behavior arises from its organization and properties at the cellular level. One way to probe these principles is to study the cell type composition in its different areas. In recent years, new data sets and tools have emerged providing the cell-type composition based on molecular identity [1,2]. However, molecular markers alone do not inform about the precise computational function of neurons, as manifested through their morphological and electrophysiological (me-) properties [3–5]. On the other hand, many established datasets propose classification schemes that are solely based on morphological and/or electrical features, while information about molecular identity is missing [6–8]. Additionally, neuronal classification based on molecular identity is not fully compatible with me-classification schemes. Nonetheless, many important results in the literature are formulated with respect to older schemes [9–12]. Looking back at such results through the prism of molecular identity would certainly provide new insights. To do so, it is desirable to define a classification incorporating morphological, electrical, and transcriptomic (i.e. molecular identity) information [5,13–15]. This requires methods to bridge the gap between data that provides the transcriptomics and other data that provides the desired me-classification schemes.

In this paper, we developed such a method allowing us to map a molecular identity to well-characterized morpho-electrical neuron types (me-types). We then took advantage of the resulting mapping to populate the mouse cortex with these me-types using the Blue Brain Mouse Cell Atlas (BBCAv2), which provides cell types densities identified by molecular markers [16]. The morpho-electrical neuron types were taken from the Blue Brain Project (BBP) morpho-electrical models database, which contains morphologies and electrophysiological recordings for detailed modeling of neurons [7]. The two main drawbacks for using detailed modeling of neurons from the BBP database are: 1) molecular identity is missing and 2) raw data used to build these models comes from juvenile rat somatosensory cortex. We thus produced a cross-species mapping between the well-established inhibitory me-types from the rat cortex and the identified molecular markers in the mouse cortex. A previous positional mapping had been derived to map the rat me-types onto rat molecular markers [17]. We improved on this previous mapping by considering electrophysiological patch-clamp recordings of cells and by taking into account variability across rat and mouse species. The BBCAv2 currently provides excitatory neuron cell types densities and inhibitory neuron densities subdivided into

PV+, SST+, VIP+ and REST densities. The REST density represents the inhibitory neurons that express markers other than PV, SST or VIP. Excitatory neurons transcriptomic classification is firstly organized per layer of origin further refined according to their projections [18]. Since the layer of origin is often already captured by m-feature based classification [7,13,14] and the electrophysiology of excitatory neurons is quite homogeneous [7,13–15], a probabilistic mapping approach for excitatory neurons is not justified. Furthermore, the morphological reconstructions we have are incomplete (i.e. no axons). We thus focused on inhibitory subtypes.

Here, we developed the method and applied it to assign a molecular identity to BBP morpho-electrical models (me-models) using a dataset from the Allen Institute for Brain Science (AIBS) [13], which provides a molecular identity to its mouse cells, bridging the gap. Since both datasets come from different species (rat and mouse), we proceeded through several normalization steps to ensure maximum alignment between the datasets to better predict common molecular identities. We then validated the approach using labels common to both datasets to verify homogeneity in cross-species resulting clusters. Finally, we used the results to perform a broader prediction: the composition of cortical brain regions in terms of established morphological and/or electrical types from recordings of marker densities. We combined the resulting mapping with molecular marker densities extracted from the BBCAv2 to infer BBP me-type densities across the whole cortex.

## Methods

### Datasets

We used two datasets in this study, one originating from the Allen Institute for Brain Science (AIBS) and the other from the Blue Brain Project (BBP).

The AIBS mouse dataset [13] served as a reference to link morpho-electrical features to molecular identity. We focused on inhibitory neurons from layer 1 to layer 6 with morphological reconstructions and electrophysiological recordings made available through the AllenSDK [19]. We collected ME data of 157 layer 2/3 to layer 6 neurons, and another set of 15 layer 1 neurons for a total of 172 neurons. We did not incorporate layer 1 neurons into the mapping process because of the low number of exemplars and because layer 1 has a specific cellular composition of types not mapped in this study [20]. Additionally, we retrieved the corresponding morphological, electrical and morpho-electrical labels from the supplementary information of the Gouwens et al. 2019 paper [13]. We defined the molecular identity (ID) as the expressed molecular marker among Lamp5, VIP, PV or SST. These four markers are known to describe a complete partition of interneurons into non-overlapping subpopulations [5,10,13,18,21]. In their supplementary information, Gouwens and colleagues provided a mapping between their native me-types classification and the 4 molecular markers Lamp5, VIP, PV and SST [13]. Thus, we could easily attribute a molecular ID to the neurons.

BBP data is both electrical and morphological. The electrical data consist of a collection of electrophysiological features (e-features). Each e-feature (e.g. action potential amplitude, interspike interval, etc.) was previously extracted from a series of electrophysiological recordings of neurons that were expertly labeled using 10 electrical types (e-types, e.g. cNAD, bIR) [7]. Thus, for each feature of a given e-type, we have a mean and a standard deviation that characterizes the distribution of values. A morphological dataset of 250 morphologies for inhibitory neurons from layer 2/3 to layer 6 allowed classification into nine morphological types (m-types) with representatives in each layer. We had an additional set of 85 morphologies classified into six m-types for layer 1. In total, we collected 335 morphologies classified into 15 inhibitory m-types (38 different m-types if we distinct them per layer).

We combined the BBP morphological and electrophysiological data to build the cell model dataset. Each cell model is a combination of a m-type with an e-type. We picked a morphology labelled with the desired m-type and assigned a set of e-features, statistically drawn from the desired e-type set of features [7]. The e-feature values of BBP cell models were randomly picked from a Gaussian distribution of values computed from the mean and standard deviation of the desired e-feature. Only m-type/e-type combinations respecting the combinations present in the digital reconstruction of neocortical microcircuitry [7] were used. We ultimately obtained 1,581 neuron models, 1,373 from layer 2/3 to layer 6 and 208 from layer 1, for which we have high-quality reconstruction and an extensive collection of electrophysiological features. In the rest of the paper, we refer to m-type, e-type and me-type labels issued from both BBP and AIBS datasets as "native" labels.

## Cross species normalization and feature extraction

Since the AIBS dataset comes from adult mouse visual cortex and the BBP dataset comes from juvenile rat somatosensory cortex we expect that most of the variance across datasets is due to differences in animal species and, to a lesser extent, differences in cortical areas and developmental variability (juvenile or adult). We thus need to normalize electrophysiological and morphological data across datasets so that the extracted features can be compared.

## Morphologies alignment

An effective normalization process needs some invariants that are used to build a framework in which data will be expressed. An invariant is a mathematical object that is conserved after transformation. In our case, the transformation is the change in animal model and/or cortical region. Since morphologies are geometrical descriptions of neurons, spatial cues that are conserved across animals and cortical areas should be used as invariants. Fortunately, in mammals, the cortex is consistently divided into layers that are homologous across species [22,23]. We thus used the layer separation of the cortex as the invariant between datasets and expressed the spatial coordinates of reconstructed morphologies in relation to these layer limits. Layer limits were derived using layers thicknesses values taken from literature [7,23,24].

First, we placed neurons in an orthonormal basis where the y axis represented cortical depth. The pia was set to $y = 0$, hence the neuron's $y$ coordinates were negative (Fig 1A*i*). We set the soma position along the x-axis to $x = 0$. We used the angles provided in Gouwens et al. supplementary information to rotate AIBS morphologies the neurons within this reference system [13]. For BBP morphologies, the original $(x, y)$ spatial coordinates were given relative to the soma. We thus shifted the neuron's $y$ coordinates to align the soma with the center of its layer of origin. The use of standard layer boundaries and the positioning at the center of a layer introduced artifacts. Some morphologies could have neurites going out of the boundaries (i.e. higher than the pia or deeper than the white matter limit).

## Neurite position normalization

To compensate for the differences in animal species we normalized the $(x, y)$ coordinates of the neurite arbor using the following set of equations:

$$x_{out} = \frac{x_{in}}{lim^{(5)}}$$

$$y_{out} = \frac{y_{in}^{(i)} - lim^{(i-1)}}{lim^{(i-1)} - lim^{(i)}} - (i-1) \; for \; i \in [1, 5],$$

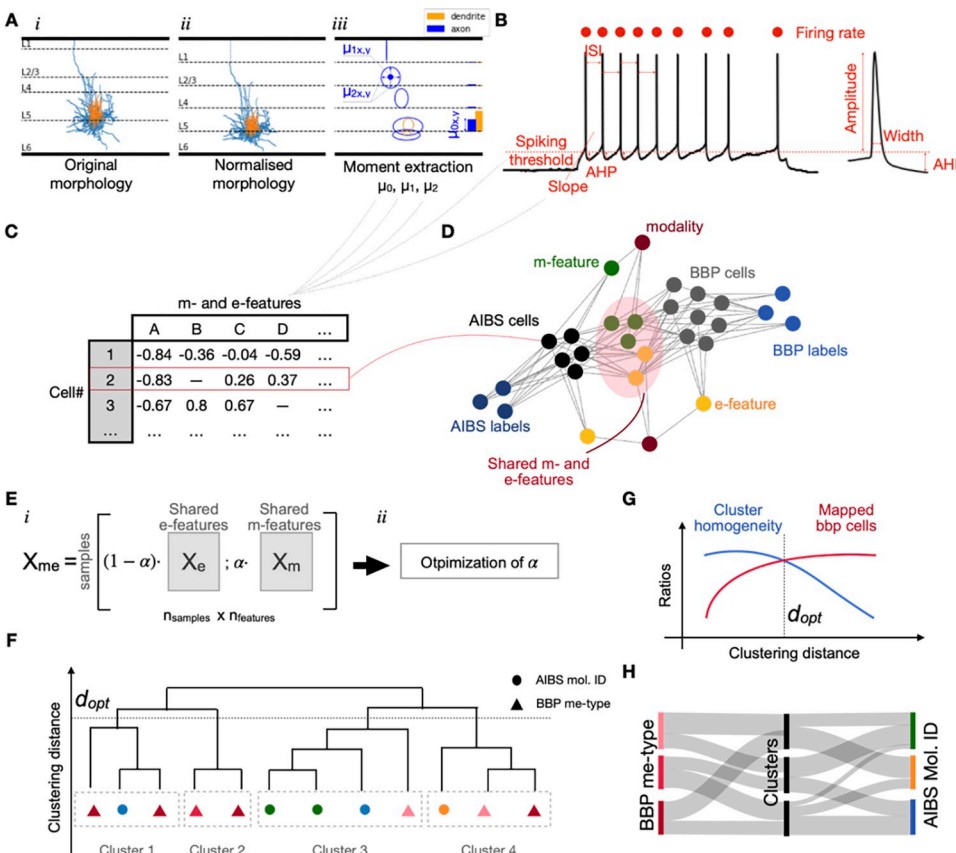

**Fig 1. Pipeline overview. A.** Morphological features extraction. Original morphologies(*i*) were realigned to be contained within cortical limits. X and Y coordinates were normalized by cortical depth and corresponding layer thickness, respectively resulting in normalized morphologies(*ii*). Density moments of order 0, 1 and 2 were extracted separately for axon and dendrites and for each layer individually(*iii*). Moments of order 0 can be related to total neurite length (bar chart on the right side of the panel) whereas moments of order 1 and 2 can be linked to mean and standard deviation of the density (ellipses). Axon in blue, dendrites in orange. **B.** Electrophysiological recordings from step currents were used to extract a collection of electrophysiological features. **C.** Extracted features were stored in tables and used to produce a knowledge graph (**D**) grouping all pieces of information that can be extracted from both datasets. Nodes are cell IDs, morpho-electrical features or other pieces of information and edges highlight the existence of a link between these pieces of information. **E.** A parameter $\alpha$ controlling the weighting between m and e features was introduced when computing the me-features matrix (*i*). We optimized $\alpha$ to find the value that yielded the most homogeneous clusters in terms of molecular IDs (*ii*). **F.** We used ward hierarchical clustering results using the optimal $\alpha$ and (**G**) we established the optimal clustering distance $d_{opt}$ as a compromise between averaged cluster homogeneity with respect to molecular IDs (blue curve) and ratio of BBP cells belonging to mixed clusters (red curve). **H.** We proceeded to probabilistic mapping using datasets native labels (see Methods) and the new assigned clusters.

where $lim^{(i)}$ is the limit of the $i^{th}$ cortical layer: $lim^{(0)}$ is the pia $y = 0$ and $lim^{(5)}$ the full cortical thickness (i.e. layer 6 bottom). Please note that layer 2 and 3 were taken together as layer 2/3 and $lim^{(2)}$ is the limit. In other words, for a given neurite arbor, the $y$ coordinates of segments located in layer 1 were normalized by layer 1 thickness, $y$ coordinates of segments in layer 2/3 were normalized by layer 2/3 thickness, etc. (Fig 1A*ii*). The resulting new coordinates for the neurite shaft thus spanned from 0 to -5 along the $y$ axis and -1 to 1 for the $x$ axis (soma centered in $x = 0$) regardless of the animal species or the brain region. In these coordinates, layer limits are always the same (layer 1 being [1, –1], layer 2/3 being [–1, –2], layer 4 being [–2, –3], layer 5 being [–3, –4] and layer 6 being [–4, –5]).

## Morphological feature extraction

After completion of the morphology normalization step, morphological features (m-features) were extracted using two different approaches. First, widely used morphometrics such as local bifurcation angles or the number of bifurcations were extracted using a package for morphological feature extraction, NeuroM [25]. All of the used NeuroM features (48 features) are listed in S2 Appendix. Second, we extended this feature collection using a parallel approach based on density moments of the axon and dendrites (Fig 1A*iii* and reference [26]. Density moments are a generalized way of extracting characteristics of a distribution such as means or standard deviations. Moments were computed using the same definition as in Snider et al. [26], S4 Appendix). It allows a straightforward comparison of morphologies but fails to encompass finer details in morphologies better captured by NeuroM features. The three first density moments were extracted for dendrites and axons separately (50 additional features).

## Electrophysiological features extraction

Electrophysiological features (e-features) such as action potential height and width were extracted from patch-clamp recordings of membrane potential responses to step current stimuli (Fig 1B). Recordings were performed using similar ACSF and ICS solutions at similar temperatures (34˚C for AIBS dataset and 35˚C for BBP dataset). We used the python package BluePyEfe developed by the Blue Brain Project [27] to extract 26 e-features (the complete list is available in S1 Appendix).

We aimed to extract me-features from similar protocols indexed by their rheobase level. The rheobase current of each cell was estimated by taking the amplitude of the smallest injected current needed to produce at least one action potential (AP). Each recording was labeled with the percentage of the rheobase current injected (from 20% to 140%). This normalizing to spiking threshold minimizes the variance due to input resistance and allows a fairer comparison between features obtained from different neurons. We then extracted electrophysiological features from all step recordings available for a given cell. In some instances, we could extract multiple values for a feature (e.g. AP amplitude in a step that triggered multiple AP). In such cases we used the averaged value for that feature. The collection of extracted features was labeled with the normalized input current amplitudes. In the end, we obtained a dictionary of features grouped by normalized step amplitudes for each neuron. However, due to differences in protocol used across datasets, not all neurons have the same set of available current amplitudes and/or features.

## Features selection for common vectorial description using a "knowledge graph"

As mentioned above, not all cells were compatible with all feature extraction methods due to the quality of morphological reconstruction or due to differences in electrophysiological protocol used. Since each cell had a different set of available features, we needed to define the best subset that is shared by the majority of cells. We computed a knowledge graph grouping all pieces of information we had about individual cells (Figs 1C and S1) using the *networkx* python package (v2.2). In this graph, each node represents one type of information such as a particular feature (e.g. mean firing frequency), a label type (e.g. BBP morphological label) but also individual cell IDs. Edges represent the accessibility of the information from a given node. For instance, the node representing a given m-type was linked to all the cell IDs labeled with the m-type. The graph was further completed with metadata nodes such as the protocol used to extract features or the animal species.

The resulting knowledge graph was used to select me-features shared by all cells from both AIBS and BBP datasets. We listed all the shortest paths in the graph linking molecular ID label nodes to individual BBP me-type nodes. These paths systematically went through an m- or e-feature and an AIBS cell id. We computed the frequency of occurrence of the me-features from the list of shortest paths. The most frequent features were used to define a common space of me-features (see the exhaustive list in S3 Appendix). We thus obtained for each neuron a vector of me-features describing the cell in the common me-feature space. Among the 722 e-features (when combining the different possible protocols) and the 98 m-features putatively available, the knowledge graph outputs 115 common me-features (26 e-features and 89 m-features, list available in S3 Appendix). Finally, the combination of the stimulus amplitude and the e-features was too restrictive to obtain common e-features for most cells. We thus allowed a degree of freedom around the stimulus amplitude when selecting e-features. For example, if in neuron A, the action potential amplitude was extracted at 120% of the rheobase but in neuron B it was extracted at 100%, we allowed the selection as long as the difference in percentage was lower than 25%. This thus ensures that e-feature values came from comparable protocols.

## Preprocessing for cross-species comparison

From the previous feature selection step, we derived four matrices: Two from the AIBS dataset $X_{e,aibs}$ and $X_{m,aibs}$ that are the e-features ($157 \times 26$) and m-features ($157 \times 89$) matrices respectively. Similarly, the other two matrices came from the BBP dataset $X_{e,bbp}$ ($1373 \times 26$) and $X_{m,bbp}$ ($1373 \times 89$). Keeping the m-features and e-features matrices separated was required for a downstream optimization step aiming at maximizing the precision of molecular ID predictions (see Methods section). The current step was designed to standardize the data to facilitate the use of clustering algorithms. To do so, features were individually preprocessed to align the mean to 0 and the variance normalized to 1 (i.e. z-scored). Each of the four matrices was preprocessed independently. Then, we concatenated the matrices of z-scored features $X'_{e,aibs}$, $X'_{m,aibs}$, $X'_{e,bbp}$ and $X'_{m,bbp}$ in a unique $X_{me,all}$ matrix as follows:

$$X_{me,all} = \begin{bmatrix} X'_{e,aibs}, & X'_{m,aibs} \\ X'_{e,bbp}, & X'_{m,bbp} \end{bmatrix}$$

The result was a $(n_{samples,aibs} + n_{samples,bbp}) \times (n_{e\text{-}features} + n_{m\text{-}features})$ matrix with $n_{samples,aibs}$ and $n_{samples,bbp}$ the number of sampled neurons in the AIBS and BBP dataset, respectively. The $n_{e\text{-}features}$ and $n_{m\text{-}features}$ are the numbers of electrophysiological and morphological features from the common me-features list.

## Evaluating dataset overlap using R-value

Before continuing the pipeline, we first sought to verify that the common me-features space we built was actually fit for cross-datasets comparisons (Fig 1A and 1B). We performed a principal component analysis (PCA) on $X_{me,all}$ and kept the components that explained more than 1% of total variance of $X_{me,all}$ (39 principal components) to obtain $X'_{me,all}$. PCAs were performed using the *PCA* built-in function from the *scikit-learn* package [28]. We evaluated the overlap of AIBS and BBP datasets using the R-value metric.

The R-value is a metric proposed by Sejong Oh to evaluate the overlap between different classes in a dataset [29]. It has been augmented by Borsos et al. to adapt it to an imbalanced problem such as the one investigated here [30]. In our case, we have 157 AIBS neurons while we have 1,373 BBP neuron models to compare with. The key notion of this metric is to look at the $k$-nearest neighbors of a given instance of one of the classes. If a significant proportion of

the neighbors belongs to another class, we consider the instance as belonging to an overlapping region. We set the parameter $\theta \in [0, k/2]$ as the threshold above which the proportion of neighbors is considered significant. If it generalizes to many points in the dataset we can conclude that classes are overlapping. In the end, we obtain a score between 0 and 1 with 0 meaning no overlap and 1, full overlap. The formal definition used for the R-value is available in S6 Appendix. In our case, we applied the R-value with $k = 400$.

We compared the dataset overlap for non-normalized features as well as normalized me-features before and after the preprocessing step. We qualitatively visualized the overlap between both datasets by plotting their first two principal components. We produced plots of the e-features, m-features and me-features space without normalization (Fig 2A), and with normalization but before (Fig 2B) and after (Fig 2C) the preprocessing.

Qualitatively, the preprocessing (i.e. z-scaling) seems to increase the overlap between AIBS and BBP dataset and a quantification using R-value (Fig 2A, insets) shows indeed an increase in both the e-feature (from 0.1 to 0.99), m-feature (from 0.79 to 0.89) and me-feature (from

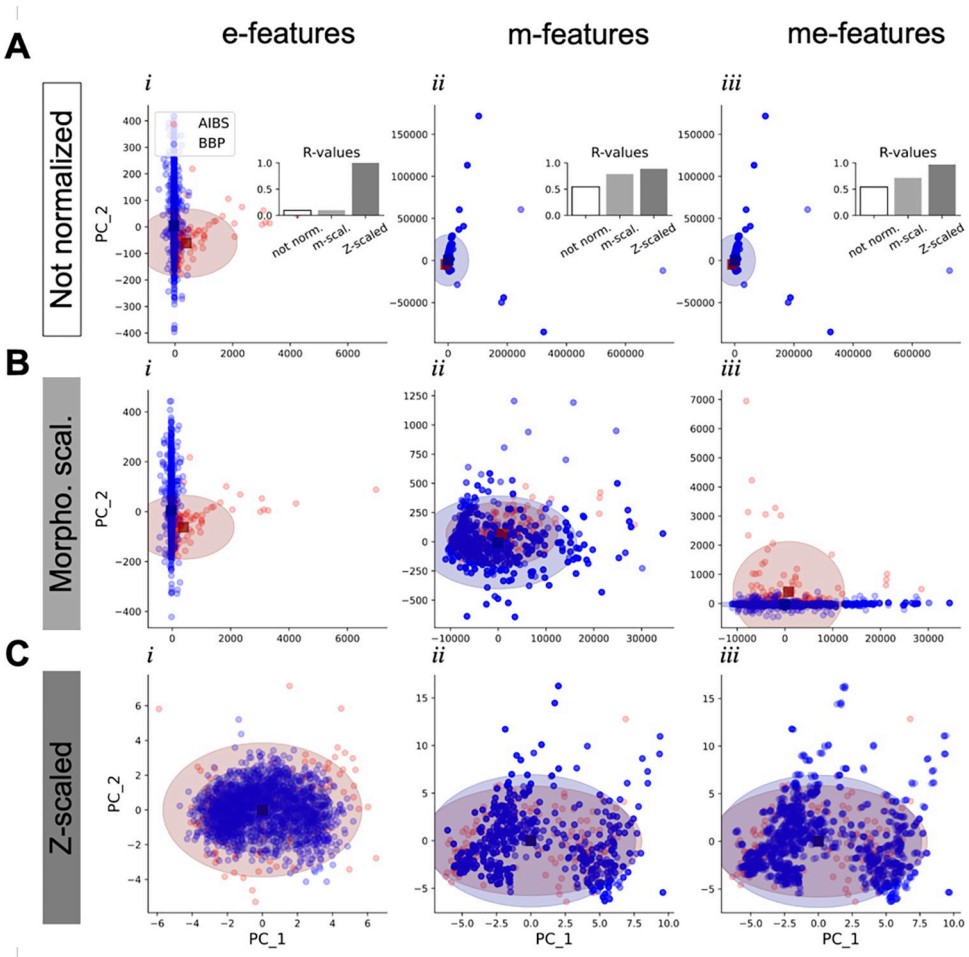

**Fig 2. Preprocessing for cross-dataset overlap optimization. A.** PCA visualization of the e-features (*i*), m-features (*ii*) and me-features (*iii*) spaces without normalization on the two first components (PC1 and PC2). AIBS cells are in red and BBP cells are in blue. Colored ellipses are centered on centroids (square dots) and radii are computed as 2 standard deviations. Insets are bar plots of R-values computed in m-features, e-features and me-features spaces without morphological scaling (white), with morphological scaling (light grey) and after (dark grey) preprocessing (Z-scaling). **B and C.** Same as above but with morphological scaling and before (B) or after (C) preprocessing (Z-scaling).

0.71 to 0.98) spaces. A normalization step before feature extraction gives good results for m-features as R-value increases from 0.54 to 0.79. However, e-features extracted from comparable step protocols (see Methods) do not produce sufficient overlap between the datasets (R-value = 0.1). There is no mathematical normalization per se on e-features, hence we observe the same PCA and R-value in Fig 3A*i* and 3B*i*. We explored different approaches to normalize extracted e-features values in more biologically relevant ways without conclusive outcomes. Thus, the preprocessing step is responsible for most of the overlap between AIBS and BBP datasets in the e-feature space.

An analysis of the e-features contribution to principal components reveals that, in the not normalized e-feature space, the first two PCs clearly capture one feature. The decay time constant after stimulation feature contributes the most to PC_1 while the doublet inter-spike interval feature contributes the most to PC_2 (S7 Fig). It appears that mouse cells scatter more along the decay-time constant while rat models scatter more along doublet inter-spike intervals (Fig 2A*i*). It is difficult to pinpoint to what degree the observation could be caused by species-specific differences as experimental divergence and algorithmic artifacts in feature attributions are likely the main causes of the differences. For instance, experimental difference in usage of series resistance compensation during recordings could lead to dataset differences in decay-time constants. On a similar note, the statistical attribution of e-features to BBP me-model might induce a bias if the e-feature distribution is not correctly modelled. This could be the case for the doublet interspike interval feature. In the not normalized morphological space, axonal and dendritic moments along the y axis in layers 5 and 6 are the most important features for components 1 and 2 (S7 Fig). However, the number of segments play a significant role as well. The latter feature is a source of bias (by the reconstructor) and is corrected by the Z-scaling. The apparent outliers in Fig 2, ii arise from higher order density moments and the size differences between rat and mouse neurons. Since the higher order moments follow square laws, a difference of 10 μm in neurite size will result in a difference of 100 in these density moments. This is why the largest morphologies from the rat appear as outliers. However, when the (x, y) coordinates are normalized relative to layer boundaries, the size differences are also eliminated (Fig 2B*ii*). Please note that Fig 2A*ii* and 2A*iii* appear identical because the use of m-features that have the largest variance masks the variance of the e-features.

Overall, this evaluation suggests that the morphological normalization process based on cortical layers boundaries is biologically relevant for cross-species comparison.

## Latent space optimization for molecular ID prediction

In their 2019 paper, Gouwens and colleagues showed that considering joint me-features instead of separated m- and e-features was the best proxy for predicting molecular ID [13]. However, the relative importance of m-features compared to e-features is not certain and is expected to be highly dependent on the available me-features. We therefore introduced an $\alpha$ weight to control this relative importance and optimize the prediction of molecular ID (Fig 3A*i*). Thus, $X_{me,all}$ matrix was computed as follows:

$$X_{me,all} = \begin{bmatrix} (1-\alpha) \cdot X'_{e,aibs}, & \alpha \cdot X'_{m,aibs} \\ (1-\alpha) \cdot X'_{e,bbp}, & \alpha \cdot X'_{m,bbp} \end{bmatrix}$$

## Hierarchical clustering analysis on common latent subspace

We proceeded to hierarchical clustering by first computing Euclidean distances between each cell pair in the common latent subspace defined by $X'_{me,all}$ (Fig 3A*ii*). We computed the linkage matrix using Ward methods as in [31,32]. We used the built-in functions *scipy* (*v* 1.5.0) *pdist*

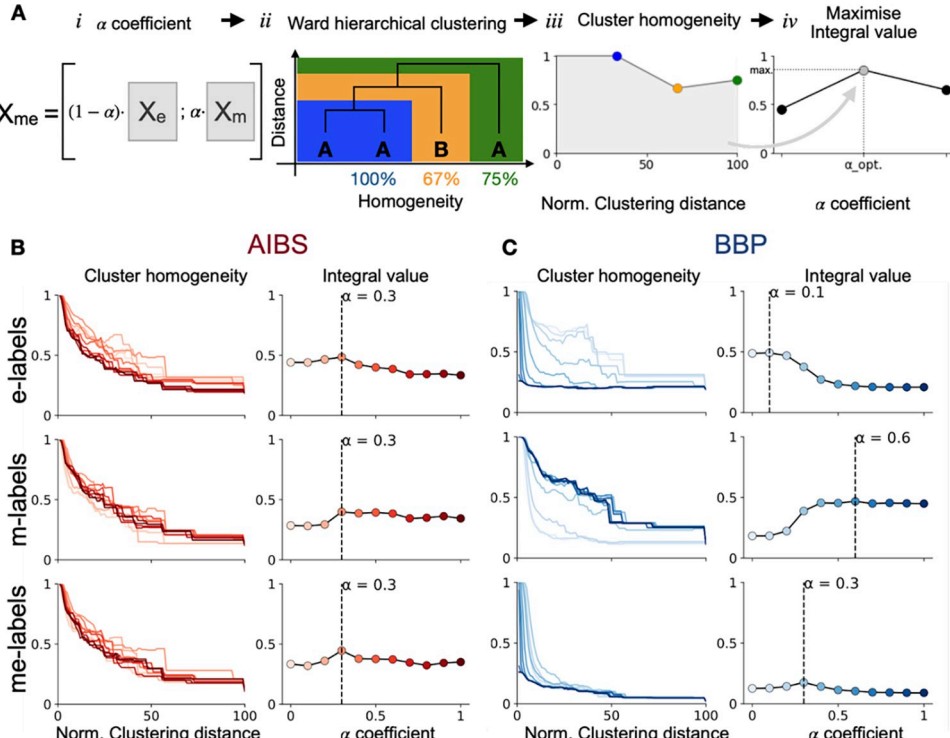

**Fig 3. Optimization of α value. A.** A parameter α controlled the weight between m and e features (*i*). We performed ward hierarchical clustering on the me-features matrix (*ii*). Cluster homogeneity with respect to a labelling system was computed depending on the hierarchical clustering distance (*iii*). We maximized the integral value to find the optimal α (*iv*). **B.** Optimization of α value using for electrical (e-labels), morphological (m-labels) and morpho-electrical (me-labels) labels of AIBS dataset. Left. Clusters average homogeneity depending on normalized clustering distance (percentage of maximal distance) for α ranging from *0* to *1*. Right. Integral values of the curves on the left panel for the different values of α. The dashed line shows the maximum integral value **C.** Same as in A but for the BBP dataset.

and *linkage* to compute distance and linkage matrices respectively. Clusters were formed by cutting the dendrogram at a given depth, later referred to as clustering distance using the *fcluster* scipy built-in function.

## α optimization and cluster homogeneity

We optimized α to maximize the cluster homogeneity of molecular ID labels. We defined the cluster homogeneity metric as the maximum percentage of the cells with the same label (e.g. molecular ID) in a given cluster (see Fig 3A*ii* and 3A*iii*). Since clusters are composed of an increasing number of cells as the clustering distance (i.e. the cut-off distance in the hierarchical tree) increases, cluster homogeneity decreases with increasing clustering distance. We optimized α by choosing the $\alpha \in [0, 1]$ that gave the highest cluster homogeneity as clustering distance increased (Fig 3A*iv*). In other words, we maximized the integral value of the cluster homogeneity versus clustering distance curve (i.e. area under the curve in Fig 3A*iii*). The optimized α for molecular ID labels was $\alpha_{opt} = 0.3$ (See Results and Fig 4A).

## Evaluation of the me-features space tuning using alpha and homogeneity metric

As a second evaluation of our pipeline we tested our ability to tune the me-features space to best predict a set of labels using the α weight and homogeneity metric (Fig 3A).

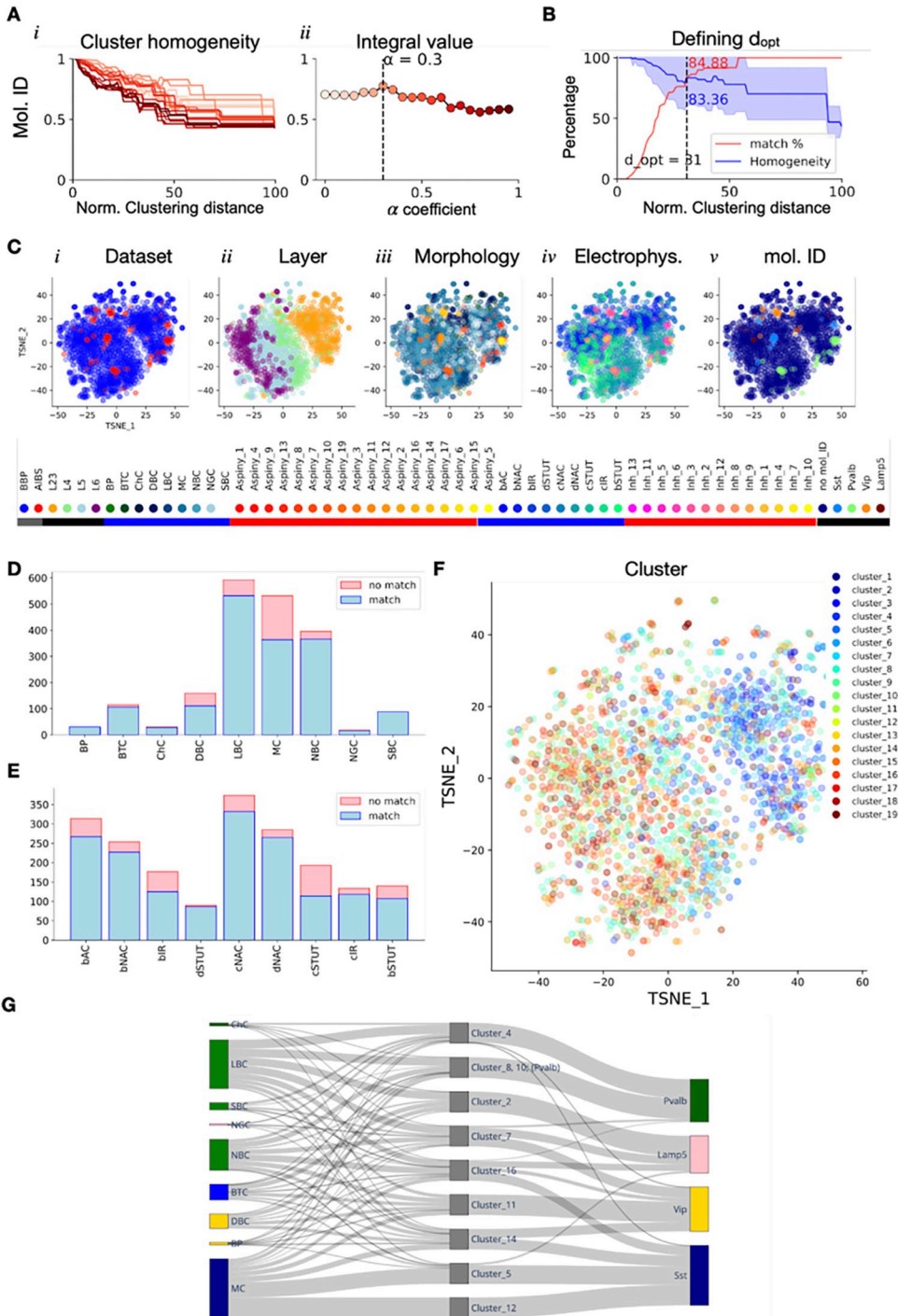

**Fig 4. Forming common clusters merging rat (BBP) and mouse (AIBS) cells. A.** optimization of $\alpha$ value using molecular ID labels for AIBS dataset. **B.** Optimal clustering distance is computed by compromising between cluster average homogeneity using marker labelling system (blue, mean +/- standard dev.) and maximizing the percentage of matched cells (red). **C.** T-SNE visualization of the optimized common subspace, data points color coded by dataset, layer labels, dataset native morphological labels, dataset native electrical labels and molecular ID. Colored bars under the legend delineate AIBS native labels (red) from BBP native labels (blue). Labels used for both datasets are underlined in black or grey. Note that the points colored in dark blue on $v$ are BBP cells for which we have no molecular ID labels. **D.** Portion of matched BBP cells grouped per BBP m-types and (**E**) e-types. **F.** T-SNE visualization of the optimized common subspace, data points color coded by common clusters. **G.** River plot showing the distribution of BBP m-types (left) and AIBS molecular IDs (right) in each common cluster (center). Clusters that mapped to one marker only

were grouped together. Clusters that mapped to BBP neurons only (cluster 1, 3, 6, 9, 13, 15, 17, 18 and 19) were excluded.

We expected the value of the optimal $\alpha$ weight to be highly dependent on the labeling system we are referring to as well as the available me-features. We thus tested $\alpha$ optimization using different native labelling systems for both BBP and AIBS cells (Fig 3). The BBP dataset clearly showed the expected results. Morphology classification (m-labels) favored higher $\alpha$ values and electrical classification (e-labels) favored lower $\alpha$ (Fig 3B). Combined mopho-electrical labels (me-labels) gave an optimal value of 0.3 between the value found with e-labels and the value found with m-labels. However, the optimal $\alpha$ obtained with all AIBS labels are identical. Nonetheless, we note that the shape of the curve is consistent with what is observed with BBP labels. Globally, the curve is decreasing with $\alpha$ for e-labels, increasing with $\alpha$ for m-labels with a threshold around 0.3 for both. The curve for me-labels is first increasing and then decreasing with $\alpha$ reaching a maximum around 0.3. The identical $\alpha$ values obtained with AIBS labels can either be explained from the relatively modest size of the AIBS dataset or from the available me-features that could not fully capture AIBS native labels leading to more noisy integral value curves.

We also looked at the effect of the $\alpha$ weight could have on R-value and the overlap of BBP and AIBS datasets (S2 Fig). The trend of R-value against $\alpha$ curve (S2 Fig) reached a maximum of 0.99 around $\alpha = 0.50$ and slowly decreased afterwards. For $\alpha \in [0, 0.5]$, we had R-values above 0.8 indicating a satisfactory overlap between AIBS and BBP datasets on this interval.

The above results confirmed that we needed to optimize $\alpha$ in regards to the labels we wanted to predict (molecular IDs in our case).

## Probabilistic mapping

We proceeded to probabilistic mapping between cells from the AIBS dataset and cells from the BBP dataset using clusters resulting from the hierarchical clustering analysis done with the optimal $\alpha$ (Fig 1E). The clustering distance was chosen to optimize both cluster homogeneities (Fig 1F) and the number of cells belonging to a mixed cluster (i.e. a cluster encompassing both BBP and AIBS cells). Since the cluster homogeneity curve decreased and the percentage of cells belonging to a mixed cluster increased with clustering distance we had to make a compromise. We thus define $d_{opt}$, the optimal clustering distance, as the distance where both curves crossed (Fig 1F).

Optimizing the clustering distance allowed us to infer a molecular ID to BBP me-models based on the molecular ID of AIBS cells belonging to the same cluster (Fig 1G). Each cell possessed a cluster label and at least another label from a native labeling system (e.g. BBP me-type for BBP cells). We used this information to derive

$$P(l_{aibs,i}|l_{bbp,j}) \forall i, j$$

with:

- $l_{aibs,i}$ the ensemble of elements from the AIBS dataset with the $i^{th}$ native label

- $l_{bbp,j}$ the ensemble of elements from the BBP dataset with the $i^{th}$ native label

  To formally define the probabilistic mapping, we introduce the following notation:

- $N$ the number of clusters resulting from clustering

- $C_i$ the $i^{th}$ cluster

- $N_{aibs}$ the number unique labels (or class) in the native labeling system of the AIBS dataset

- $N_{bbp}$ the number unique labels (or class) in the native labeling system of the BBP dataset

We then estimated the probability of a cell from test dataset to belong to $l_{aibs,i}$ knowing it belongs to $l_{bbp,j}$ as:

$$P(l_{aibs,i}|l_{bbp,j}) = \sum_{k=0}^{N} P(l_{aibs,i}|C_k) \cdot P(C_k|l_{bbp,j})$$

With $P(l_{aibs,i}|C_k) = \frac{card(l_{aibs,i} \cap C_k)}{card(\cup_m^{N_{aibs}} l_{aibs,m} \cap C_k)}$ and $P(C_k|l_{bbp,j}) = \frac{card(l_{bbp,j} \cap C_k)}{card(l_{bbp,j})}$, where card(x) is the cardinal (i.e. the number of elements) of the set x.

Applying this formula comes with the following prerequisite:

When taken individually, each of $l_{aibs}$, $l_{bbp}$ and $C$ ensembles are complete, non-overlapping ensembles of the cell-type ensemble. Completeness means that one cell sample had only one label from each of the $l_{aibs}$, $l_{bbp}$ and $C$ ensembles. We forced this condition by excluding the cells belonging to dataset exclusive clusters (e.g. clusters with only BBP cells).

We proceeded similarly for

$$P(l_{bbp,j}|l_{aibs,i}) = \sum_{k=0}^{k} P(l_{bbp,j}|C_k) \cdot P(C_k|l_{aibs,i})$$

and applied it to derive $P(\textit{me-type}_{bbp}|\textit{marker}_{aibs})$.

## T-distributed stochastic neighbor embedding visualization

We visualized the optimized subspace using T-distributed stochastic neighbor embedding (T-SNE). We used the *scikit-learn* package [28] and the *TSNE* built-in function with the following parameters: n_components = 2, perplexity = 30.0, early_exaggeration = 12.0, learning_rate = 200, n_iter = 1000, n_iter_without_progress = 300, min_grad_norm = 1e-07, metric = 'euclidean', init = 'random', verbose = 0, random_state = None, method = 'barnes_hut' and angle = 0.5.

## Labels common to both datasets used for pipeline validation

We validated the whole pipeline using common labeling systems based on morphology or the layer of origin.

## Common morphological labels

Both AIBS and BBP m-types were related to broader morphological types often used in the literature (e.g. basket cells, Martinotti cells). The grouping was done according to the morphological description of me-types referenced in Gouwens et al. supplementary data [13]. This comes with the cost of merging together several m-types in their respective native labeling systems (i.e. AIBS or BBP), hence decreasing resolution, but necessary to acquire common morphological labels (S1 Table).

## Common layer labels

Both the Gouwens et al. and BBP datasets provided layer-specific labels [13]. We merged the Gouwens et al. dataset L6a and L6b labels into one L6 label and obtained five different common labels: L1, L2/3, L4, L5 and L6.

## Marker to me-types densities conversion

$P(me\text{-}type_{bbp}|marker_{aibs})$ was derived using the probabilistic approach described above (see S6 Appendix for further details). We subdivided the marker labels from Gouwens dataset layer-wise aiming at a finer mapping.

The BBCAv2 provides densities for PV, SST and VIP expressing cells (hence, *Densities (markers)*) since they are commonly-used markers defining non-overlapping interneuron sub-types [18,21]. The REST density for inhibitory cells expressing none of the previously mentioned markers was also given [16], S5 Appendix). To include all interneurons densities, we assumed neurons from REST densities as being Lamp5 positive. We expected using the four markers PV, SST, VIP and Lamp5 would give a reasonable estimate of interneuron subpopulations. We computed densities of the $i^{th}$ BBP cortical inhibitory me-type using:

$$Densities(me-type^{(i)}) = \sum_{j} P(me-type^{(i)}|marker^{(j)}) * Densities(marker^{(j)})$$

## Results

### Pipeline overview

The main goal of this study was to populate a digitalized mouse cortex in a cell atlas with morpho-electrical models from the BBP. To that end we used molecularly defined cell-type densities from the BBCAv2 as input and sought to convert them into well characterized me-types for which we have detailed models. Such conversion required the development of a method able to assign a molecular ID to the desired me-types. The core idea of the method was to learn from a reference dataset [13] how to link molecular ID to a set of morpho-electrical features and apply this extracted knowledge to BBP me-models. We focused on inhibitory neurons since biological markers defining their subclasses have been well established in the neocortex (e.g. PV, VIP, SST). On the other hand, the consensus defining molecular subclasses of excitatory neurons is less clear and appears to be based on the cortical layer of origin and axonal projections [5,18,21]. Since the BBP excitatory me-type labels incorporate the layer of origin, the mapping is trivial at the coarse level. The pipeline for molecular ID assignment is composed of five major steps. The first step consists of morpho-electrical features (me-features) extraction with an embedded normalization to minimize variability due to differences in animal species or cortical region of origin (Fig 1A and 1B). We used cortical layer boundaries for morphological normalization and estimates of rheobase current to extract e-features from comparable protocols (see Methods). Step two is a feature selection module based on a knowledge graph that outputs common me-features between both datasets necessary to build a common me-features space (Fig 1C). Then the common me-features space is optimized (step three) to maximize precision of molecular identities predictions (Fig 1D). The fourth step consists in cluster formation using Ward hierarchical clustering (Fig 1E and 1F) and the final step is a probabilistic mapping (Fig 1G) to assign molecular ID to neurons from the incomplete dataset. Since inhibitory neurons from layer 1 have been shown to be quite distinct from inhibitory neurons of other layers [20], they were excluded from steps three to five (see the next two paragraphs).

### Forming cross dataset clusters

We applied the pipeline to predict the molecular ID (i.e. marker expression, see Methods) of BBP me-models. The BBP models were built by taking a morphology with the desired m-label and attaching a set of e-features statistically drawn from a collection of recordings with the desired e-label. We thus optimized $\alpha$ for molecular ID labels focusing on the Gouwens dataset

for which we had molecular IDs. It has been previously established that considering morpho-logical and electrophysiological properties of cells together better predicts their molecular IDs [13–15]. However, it is not known in advance what importance to give to morphological properties relative to electrophysiological properties. Optimization of $\alpha$ with molecular IDs gave a $\alpha_{opt} = 0.3$ close to the optimal $\alpha$ value obtained with native me-labels for BBP and AIBS datasets (Fig 4A). A T-SNE visualization of the optimised me-feature subspace showed that both data sets are well mixed although, locally, we observed species-specific clusters (Fig 4C). Color coding data points per layers, dataset native m-types or native e-types shows that the data is organized per layer first, then by electrophysiological and morphological types within the layers. In turn, color coding per molecular ID shows that neurons with the same molecular ID tend to cluster together. Cross dataset clusters were obtained by cutting the hierarchical tree at $d_{opt} = 31\%$ of maximal clustering distance (See Methods). At $d_{opt}$, approximately 85% of cells belonged to merged clusters while averaged cluster homogeneity was still around 83% (Fig 4B). We obtained 19 clusters with this approach (Fig 4F) with 76% of all BBP cells from all m- and e-types belonging to a cluster merging cells from both datasets (Fig 4C). Clusters 1, 3, 6, 9, 13, 15, 17, 18 and cluster 19 were exclusively composed of BBP cells (excluded from the river plot, Fig 4G), while none of them contained AIBS cells only. Formation of dataset exclusive clusters could be explained by the dramatic difference between dataset sizes but might also be a hint towards species specific cell-types.

Finally, we estimated cluster composition in terms of BBP me-models and AIBS (Fig 4G). Each BBP and AIBS neuron had a native label and a shared cluster label. Both labeling systems can be used interchangeably but for clarity, we used BBP m-type instead of me-types as BBP native labels (only 9 m-types compared to more than 80 me-types). As for the AIBS dataset, we used the expressed marker. For both the BBP and AIBS datasets we computed the proportion of respective intrinsic labels composing a cluster. These proportions were used for the river plot in Fig 4G.

## Combining mapping and marker densities from the blue brain mouse cell Atlas v2 to populate cortex with BBP me-models

The output from the previously described pipeline can be used to estimate densities of BBP me-types based on molecular marker densities using a probabilistic approach as defined by:

$$Densities(me - type^{(i)}) = \sum_{j} P(me - type^{(i)}|marker^{(j)}) * Densities(marker^{(j)})$$

with,

$$P(me - type^{(i)}|marker^{(j)}) = \sum_{k} P(me - type^{(i)}|cluster^{(k)}) \cdot P(cluster^{(k)}|marker^{(j)})$$

where $P(me\text{-}type^{(i)}|cluster^{(k)})$ and $P(cluster^{(i)}|marker^{(j)})$ are derived from the pipeline results (Fig 5A, Methods). On the other hand, $Densities(marker^{(j)})$ is provided by the updated Blue Brain Mouse Cell Atlas [16].

Thus, multiplying these matrices together estimates the densities of BBP me-types based on densities of molecular markers. An example of the obtained estimates but using BBP m-types for clarity, is given in Fig 5B and 5C. For completeness, we incorporated L1 BBP m-types using densities of glutamic acid decarboxylase or GAD positive cells provided by the BBCAv2. Thus, relative proportions of L1 m-types were not given by the mapping but came from their relative proportions in the BBP dataset instead. For density profiles (Fig 5C), we focused on two cortical regions: Somatosensory cortex, hindlimb area, and the visual primary cortex. A

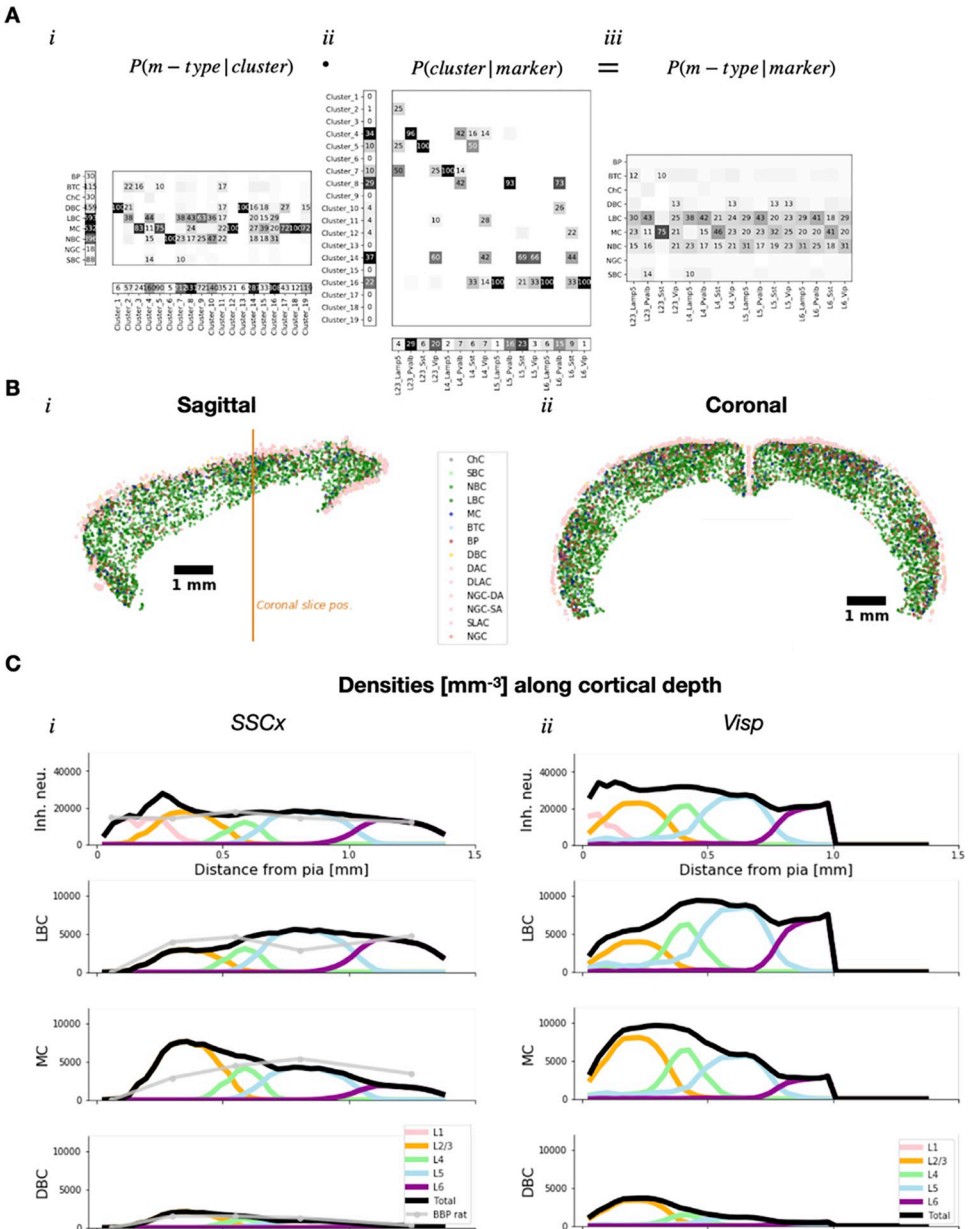

**Fig 5. Estimates of BBP m-type densities using BBCAv2. A.** Probability matrices to illustrate the equation used for m-type densities estimates. For clarity, $P(m\text{-}type|cluster)$ instead of $P(me\text{-}type|cluster)$ is shown in (*i*). Similarly, $P(m\text{-}type|marker)$ is shown in (*iii*) instead of $P(me\text{-}type|marker)$. Probability values higher than 0.1 are displayed as percentages in matrices. $P(m\text{-}type|cluster)$ was computed on BBP dataset in (*i*). The number of BBP me-models in each cluster is given in the bottom row, while the number of cells of each m-type is given in the first column. $P(cluster|marker)$ was similarly computed on AIBS dataset in (*ii*). The number of AIBS cells in each cluster are given in the first column, while the number of cells for each AIBS molecular ID is given in the bottom row. **B.** Typical distribution of m-types in mouse cortex. A populator algorithm took as input cortical densities of m-types in 25 μm wide voxels and assigned spatial coordinates to neurons in a random fashion within the same voxel. Sagittal and coronal view of inhibitory neuron distributions are color-coded by m-types. Orange vertical bar in *i* shows the position of the coronal slice. **C.** BBP m-types densities along cortical depth computed by applying the $P(m\text{-}type|marker)$ matrix to marker densities from the BBCAv2. Two different cortical areas are displayed: primary somatosensory cortex, hindlimb area (SSCx, *i*) and primary area in visual cortex (Visp, *ii*). Densities of all inhibitory neurons are shown in the first row and only LBC, MC and DBC m-types are shown for clarity (all m-types are available in S3 Fig). Other m-types and me-types are similarly computed. Contributions per layer are shown as colored lines while the total is displayed in black. For comparison in somatosensory cortex, data from rat neocortical microcircuitry reconstruction are shown in light grey.

bias toward an over-representation of Martinotti cells (MC), nested and large basket cells (NBC and LBC) seems to be present. This is most likely due to the many examples of these cell types in the BBP dataset. Another important point to notice is that some probabilities of $P$ (cluster$^{(i)}$|marker$^{(j)}$) are derived from very few representatives from a given molecular ID (e.g. *L5 Lamp5*). With these exceptions in mind, the mapping seems reasonable with $P$(*m-type*|*L5 PV*) being maximum for LBC (0.43) or $P$(*m-type*|*L2/3 SST*) being maximum for MC (0.75). Overall, predicted densities seem realistic especially when compared with previous data from BBP digital reconstruction of rat neocortical microcircuit (Figs 5C and S3). Finally, we also explored the variation of densities across cortical regions (S4 Fig). We could observe again the overrepresentation of LBC, NBC and MC m-types across all regions (S4A Fig). A quick analysis of how m-type densities in a given subregion diverged from their mean over the whole isocortex showed a relatively smooth distribution over the cortex (S4B Fig). We computed divergence ratios to see how much densities of a given region were diverging from the mean over the whole isocortex (Methods). Divergence ratios spanned from -60% to 120% with maximal positive divergences observed in layer 6 of the orbital ventrolateral area for BTC, DBC and MC m-types. It seems that changes in divergence ratios correlate well with layer labels. This is expected as m-types variations come from the marker densities variation from the BBCAv2 and we considered the marker densities from different layers separately when we computed $P$ (*me-type*$^{(i)}$|marker$^{(j)}$) The obtained me-type densities have been incorporated to the BBCAv2 website for public availability (https://bbp.epfl.ch/nexus/cell-atlas/).

## Validation of the cross-dataset clustering pipeline

We submitted our pipeline to two types of validations. First, a validation for molecular type prediction and second, a validation for cross-species prediction.

The first kind of validation means that we want to test how good our approach is to predict the molecular ID of a neuron solely based on the available me-features. Since we had no information about the molecular ID of BBP me-models we restricted ourselves to the AIBS dataset only. We split the AIBS dataset into 10 subsets of comparable sizes and proceeded to perform a 10-fold cross-validation of the pipeline's prediction. We did the $\alpha$ optimization step using 9 subsets grouped together and predicted the molecular ID of cells from the last subset by assigning them to the most frequent molecular ID of their cluster. For clarity, the hierarchical tree was computed using the complete dataset for every tested $\alpha$ while the cluster homogeneity score was computed with labels from the nine grouped subsets used as training dataset. We repeated this for the 10 possible combinations of the 10 subsets. Results show promising weighted precision and accuracy (85% +/- 13% and 85% +/- 10% respectively, see Table 1) scores well above chance levels. Precision measures the rate of the same prediction in a class. For instance, it tells how many of all the Pvalb cells were predicted with the same label. Then we averaged obtained precisions over all classes but weighted them by the number of elements in a class giving the global weighted precision. Accuracy on the other hand measures the rate of correct predictions (e.g. How many of the predicted PV labels are actual PV$^+$ cells).

For the second kind of validation we used both common labeling systems, i.e. labels that can be used to describe neurons from both the BBP and AIBS datasets. As defined in the Methods section, we used two common labeling systems: one based on the cortical layer where the soma resides and another based on consensual morphological types.

First, we assessed the performance of the pipeline at predicting common m-type and common layer labels within a dataset. We then proceeded to a 10-fold cross-validation for each of the following cases: common m-type labels on Gouwens et al. dataset, common layer labels on Gouwens et al. dataset, common m-type labels on BBP dataset and common layer labels on

**Table 1. Validation experiments results.** Section A: 10-fold cross-validation on the Gouwens et al. dataset using molecular markers, common m-types and common layer labels. 10-fold cross-validation on BBP dataset using common m-types and common layer labels only. Mean (standard deviation) are shown except for the chance level column where it is mean (median). Section B: Cross dataset validation on common m-types and on common layer labels, respectively.

| | | Training dataset | Alpha | Precision (weighted) | Accuracy | Chance Mean (median) | Ratio of dataset matched |
|---|---|---|---|---|---|---|---|
| **A. 10-fold cross validation** | **Molecular IDs** | Gouwens et al. | 0.38 (0.04) | 0.85 (0.13) | 0.85 (0.1) | 0.25 (0.24) | 1.0 (0.0) |
| | **Common m-types** | Gouwens et al. | 0.4 (0.0) | 0.89 (0.09) | 0.79 (0.1) | 0.2 (0.2) | 0.96 (0.07) |
| | | BBP | 0.58 (0.1) | 0.81 (0.02) | 0.81 (0.02) | 0.2 (0.16) | 0.98 (0.02) |
| | **Common layers** | Gouwens et al. | 0.74 (0.1) | 0.76 (0.17) | 0.69 (0.13) | 0.25 (0.24) | 0.95 (0.05) |
| | | BBP | 0.74 (0.14) | 0.95 (0.02) | 0.95 (0.02) | 0.25 (0.25) | 0.99 (0.01) |
| **B. Cross-dataset** | **Common m-types** | Gouwens et al. | 0.3 | 0.41 | 0.4 | 0.2 (0.2) | 0.72 |
| | | BBP | 0.6 | 0.83 | 0.78 | 0.2 (0.16) | 0.84 |
| | **Common layers** | Gouwens et al. | 0.7 | 0.6 | 0.55 | 0.25 (0.23) | 0.73 |
| | | BBP | 0.5 | 0.59 | 0.47 | 0.25 (0.25) | 1.0 |

BBP dataset. Precision and accuracy scores were very satisfactory (Table 1) showing that the pipeline is able to correctly predict common m-types and common layers when training on a dataset from the same species.

We tested the ability of the pipeline to do cross-species prediction. This process consisted in running the full pipeline but with a common labelling system (Table 1). We did the training on one dataset and the prediction on the other dataset. In other words, the $\alpha$ optimization step was performed on the AIBS dataset (or on the BBP dataset) and cells from the BBP dataset (respectively, from the AIBS) were assigned a label using the most frequent label of the cluster they belonged to. For clarity, the hierarchical tree was computed using all data points from both BBP and AIBS datasets but the cluster homogeneity score was computed with labels from one dataset only. A "no-match" case in which the cluster is 100% composed of AIBS (or BBP) cells could also be observed. In such cases, data points were excluded. As a result, only a subset of the tested dataset was predicted. For instance, for common m-labels prediction of BBP cells when trained on the Gouwens dataset, we predicted 72% of the BBP dataset with a precision of 41% and an accuracy of 40% (Table 1). Conversely, when we trained with the BBP dataset and predicted common m-labels of AIBS cells, we obtained a precision and accuracy score of 83% and 78% respectively, while 84% of the AIBS dataset had a predicted label.

Interestingly, when training on the AIBS dataset the $\alpha$ value is lower ($\alpha = 0.3$) compared to when the BBP dataset was used for training ($\alpha = 0.6$). While we would expect a $\alpha > 0.3$ for predicted morphological labels, this might partly be explained by the large difference in dataset sizes or due to residual electrophysiological information in the classification as we will develop in the discussion. This difference in $\alpha$ values could also be due to the imbalance in the reference dataset labels. For instance, the BBP dataset has a sizeable amount of basket cells (S5 Fig) potentially driving the optimal $\alpha$ value towards a more systematic prediction of basket cells.

We proceeded similarly with layer labels (Table 1). However, the precision and accuracy scores were lower (Table 1) when compared to scores obtained using common m-labels for the case with training on BBP dataset. This could be due to the artificial placement in layers of BBP neuron morphologies (see Methods).

Since only morphologically related common labels were available for cross dataset validation, we have little information about the ability of the pipeline for electrophysiology-related cross-dataset predictions. We nonetheless explored the results obtained when applying the pipeline with electrophysiological labels. In the provided description of their me-types, Gouwens et al. reported electrophysiological phenotypes that were regularly used in literature (e.g.

with fast spiking neurons) [13]. We thus could easily group Gouwens et al. me-types into four "common" e-types: fast spiking (FS), regular spiking (RS), irregular spiking (IR) and adapting (Adapt). We ran the pipeline optimizing $\alpha$ for these "common" e-types to output the probabilistic mapping $P(common\ e\text{-}type|BBP\ e\text{-}type)$ (S6A Fig, see Methods). This mapping clearly suggests that most of the dNAC, bSTUT, cSTUT and dSTUT labeled neurons map to FS neurons. All the other labels map to RS, IR and Adapt but not to FS neurons. To push the analysis further, we output the probabilistic mapping $P(marker|BBP\ common\ me\text{-}type)$ (S6B Fig). Common me-types were defined as common m-type with common e-type. For the BBP dataset, dNAC, bSTUT, cSTUT and dSTUT labels were grouped under the FS common e-types while the other labels were classified as non-fast spiking (nFS). The resulting mapping suggests that, using the same morphology but matching it to a different set of e-features, either FS or nFS, could change the molecular ID of the resulting neuron model. For instance, an MC morphology grouped with nFS e-features would most likely map to SST positive neurons while if the morphology is grouped to FS e-features it will be mapped to PV positive neurons. These results suggest that considering electrophysiological and morphological dimensions separately when building neuron models could increase noise and confusion in the mapping.

Overall, these results suggest that the pipeline is well suited for molecular ID prediction and is able to do cross-dataset predictions while more refinement of features normalizations would likely increase the precision of predictions.

## Discussion

In this paper we populated a digitalized mouse cortex with well characterized inhibitory me-types from the BBP neuron model database, thereby paving the path for detailed modeling of this whole brain region. In the process we created a common framework to effectively compare mouse and rat inhibitory cells. This framework was subsequently used to map partially randomized neuron models from the rat to mouse neurons from a recent AIBS dataset [13] in order to extrapolate missing molecular IDs. The resulting probabilistic mapping served as a convertor to estimate me-type densities using molecular marker densities from the BBCAv2 as input. Results showed relatively comparable distributions across the cortical areas (Figs 5 and S4).

A major strength of the approach presented here is the ability to accommodate data coming from different species. To support the validity of the approach, we first relied on previous studies pointing out the conserved nature of some transcriptomically defined cell types across different vertebrate animals [33,34]. Second, some functionally defined cell-types that are observed in both rat and mouse species exhibit comparable functional features. For instance, PV+ basket cells show fast spiking profiles in monkey, mouse and rat cortex [7,13–15,21,35]. Similarly, Martinotti cells show similar morphologies in both species with axons reaching layer one to make synapse with apical dendrites pyramidal neurons. They also exhibit regular spiking behavior [13–15,36,37]. These studies clearly suggest that transcriptional and functional parallels can be drawn between closely related species while being aware that species-specific as well as cortical region-specific variability and exclusivity is to be expected. Normalization processes made possible the expression of me-features in a common framework necessary for cross-species comparison. Doing so required invariant values across species such as layer delineation for morphologies and rheobase current for electrophysiology (see Methods and Fig 1B). Although the normalization could be improved for e-features, it showed promising results for m-features (Fig 2). Further refinements of the normalization process improving dataset compatibility would help minimize mapping uncertainties. Additionally, we showed that a well-identified morphology grouped with different e-types could be associated with

different molecular markers (S6 Fig). These results confirm that considering morphology and electrophysiology as two fully independent dimensions might increase confusion in the mapping process [13–15]. However, with further exploration, these normalization methods could help to adapt data gathered in the rat and utilize them for building a model of the mouse cortex.

Optimization of the latent space to maximize prediction of molecular IDs confirmed that considering both m- and e-features together makes more sense when trying to predict molecular ID. Indeed, the $\alpha_{opt}$ obtained with molecular ID labels had a value close to the $\alpha_{opt}$ obtained with me-type labels (Fig 3). These results agree with multiple previous studies searching for links between molecular IDs and morpho-electrical description of neurons [13–15]. However, the latent space optimization step presented here does not inform on the exact features that are important for molecular ID predictions. One can expect that not all of the extracted features are of the same relevance when it comes to predicting the molecular ID of a neuron. A refined version of the optimization that we are currently exploring considers having individual weights for each principal component instead of only one. Thus, we would have a list of $\alpha_i$, one for each principal component, that could be optimized using a multivariate optimization algorithm with a loss function inspired from cluster homogeneity. As a result, we would expect me-features from principal components with high $\alpha_i$ values to be more relevant for molecular ID prediction. This could also improve the $\alpha_{opt}$ result for the Gouwens dataset with native labels (Fig 3B). Another possible explanation for these results is that the native AIBS e-, m-, me-labels are more congruent with each other compared to the BBP native labels. The congruence of the AIBS labels is suggested in their study and is further supported, by the fact that their native e-, m- and me- labels usually correlate well with molecular labels [13].

To form clusters in the optimized latent space, we showed there was a trade-off between the molecular ID precision, via cluster homogeneity, and the number of mapped BBP models (Fig 4B). Therefore, we made a compromise between cluster homogeneity and BBP cell mapping percentage. As an alternative, one can consider increasing cluster homogeneity by choosing a smaller clustering distance, hence preferring molecular ID precision over the number of mapped BBP cells. The final choice should depend on the goal and should be at the user's discretion. On the other hand, this trade-off can leave some room to identify species-specific cell types. It is highly likely that strictly rat-specific cells would not cluster well with their mouse equivalents (if they exist), which makes our method a strong contender compared to other supervised methods where unique cell clusters might be forced to cluster together in a less specific group. For example, three clusters were composed of BBP cells only, which could suggest rat-specific cell types. However, with more than 1,300 BBP models and as few as 150 neurons from the Gouwens dataset, we are dealing with datasets of different sizes. Due to datasets size differences, it is hard to discern if we are in a rat exclusive cell-type case or, more likely, if there are not enough neurons in the Gouwens dataset to map these clusters (Fig 5A*ii*). It is also possible that these species-specific clusters could result from a combination of a morphology and e-features that does not actually exist in the biological population. We showed in S6 Fig that the BBP me-type construction process could potentially assign incompatible markers to m-types depending on the e-type attached to it. Furthermore, one cluster, Cluster 2, contains only one AIBS cell to infer molecular IDs for the BBP neurons. Consequently, the inferred molecular IDs for BBP cells in this cluster should be regarded with caution. The use of a larger reference dataset balancing the larger BBP dataset could help reduce these mapping uncertainties. For instance, new Patch-seq datasets combining morphological reconstruction, electrophysiological recordings and scRNAseq for individual neurons are becoming increasingly available [14,15,38,39]. Future inclusion of such datasets would improve the precision of mapping by equilibrating the number of mouse cells compared to rat cells.

The mapping results also points towards the presence of morpho-electrical variability within a molecularly defined cell-type, as has been clearly demonstrated in recent studies [14,15]. Since the mapping process yields optimized results with a number of clusters much greater than the four molecular markers used in the mapping, additional variability is expected. Otherwise, we would obtain an optimized mapping with a number of clusters close to the number of molecular markers.

Finally, we showed that using a probabilistic mapping approach, we were able to estimate densities of well characterized me-type across cortical areas. However, this approach assumes that the cell repartition in labels is unbiased (i.e. it reflects what we could observe naturally). We know that it is not the case for the AIBS dataset, as some specific driver lines have been used to boost up the probability of observing some types of neurons such as chandelier cells with the Vipr2-IRES2-Cre driver line. To correct for this, we explored using only non-overlapping driver lines for our probabilistic mapping when estimating m(e)-type densities. However, choosing non-overlapping driver lines reduced the already modest number of AIBS cells. In addition, the resulting dataset might still be biased due to experimental methods such as the accessibility of cells for patching, for instance. We thus chose to proceed with the complete AIBS dataset. The BBP dataset could present a similar experimental bias with over-representation of LBC or MC for instance (S5 Fig). However, it remains difficult to estimate to which proportion the numbers from the BBP dataset are due to natural occurrence or experimental biases. The obtained m-type densities showed similar distributions across cortical regions for a given m-type (Figs 5C and S4). The variation of me-type densities across cortical regions are necessarily rooted in the input marker densities since we linearly transformed these inputs using a unique probability matrix $P(me\text{-}type|marker)$. It should also be noted that this matrix was derived using cells from juvenile rat somatosensory cortex and adult mouse visual primary cortex. Including cells from other brain areas and comparable development stages could improve precision of me-types predictions. Nonetheless, these results provide a first estimate of inhibitory me-types composition for the whole cortex enabling more precise modelling of this brain region in the future. In addition, the next iteration of the BBCA is likely to provide an atlas of more finely defined t-types. We plan to couple this upgraded version with a more refined mapping of finely defined t-types to BBP models using Patchseq datasets [14,15].

These results should be nuanced by the outcomes of our validation tests. Molecular ID prediction gave very good precision and accuracy scores, supporting the validity of the approach (Table 1). Unfortunately, for cross dataset validations, only morphology-related labels were available. We observed averaged scores with an unexpected low $\alpha$ for common morphological labels prediction (Table 1). These results might be partly explained by the lack of objective, consensual morphological classification in the literature. However, the better results observed when training on the BBP dataset suggest dataset size differences could also play a role in the obtained validation scores. It is possible that the AIBS dataset is too small to fully encompass all morphological diversities, potentially resulting in inaccurate and imprecise predictions. Concomitantly, the fact that the native AIBS e-, m- and me-labels are congruent with each-other and that we can derive common m-types from these native labels could imply that there is residual electrophysiological information in these labels for the AIBS dataset. This residual information could bias the pipeline when training on the AIBS dataset with common m-type labels, leading to a lower $\alpha$ and ultimately to a decrease of the mapping efficiency when assigning a common m-type to BBP models. Nonetheless, we found the results satisfying in the context of molecular IDs for cross-dataset mapping (Table 1).

A valid technical criticism about the pipeline is that all data points are considered when doing the hierarchical clustering, even the points for which we don't have labels. For instance,

when we optimized our latent space for molecular IDs prediction, we proceeded to hierarchical clustering using both AIBS cells (with known molecular IDs) and BBP cells (with unknown molecular IDs). Thus, we might influence the actual value of alpha to our advantage to predict the unknown labels. One could argue nonetheless that hierarchical clustering is used here as a deterministic mathematical tool to probe the common space structure. The alpha optimization is then performed on this projection of the subspace. Hence, the actual optimization uses only the data points with known labels.

It is possible to generalize the mapping approach presented here as long as a common framework can be defined to allow comparisons between datasets. The optimization of the subspace resulting from this common framework will always find the best $\alpha$ based on the available features. Building an extensive cell-type—based knowledge graph linking multiple datasets will be a useful resource to establish such frameworks. This initiative is already under process both at BBP, AIBS, and in other groups. In addition, incorporating multiple datasets will not only increase the range of information types that we can infer but also the precision and confidence in the inferred information. In the future, Patch-seq dataset integration could possibly be generalized to infer even specific me-features solely on gene expression profiles [40,41]. The ultimate goal would be to make precise predictions about neuronal types of uncharted brain areas from very sparse data. An essential key to this problem is the definition of a brain area independent framework in which we can express morphological and electrophysiological features. For other regions than the cortex, or even other species, morphology normalization using layer delineation might not be well adapted, thus other "invariants" must be found. In addition, the low R-value observed with normalized e-features (Fig 2) suggest that electrophysiology normalization using rheobase current should be improved. If the newly defined framework is general enough, it could even be conserved across species (e.g. mouse and rat) also enabling cross-species predictions.

We present here a first step towards what we think is an essential task: to develop and refine algorithms that infer missing composition knowledge from what is already known. Such algorithms will not only help to draw parallels across different animal species but also extend our comprehension of less studied brain areas, thus facilitating the process of building a biologically detailed model of the mouse brain.

## Supporting information

**S1 Appendix. Electrophysiological features list.**
(DOCX)

**S2 Appendix. NeuroM features list.**
(DOCX)

**S3 Appendix. Common me-features list.**
(DOCX)

**S4 Appendix. Neurite density moments extraction.**
(DOCX)

**S5 Appendix. Brief description of maker densities extraction.**
(DOCX)

**S6 Appendix. Formal definition of R-value.**
(DOCX)

**S1 Table. Correspondences between m-types or me-types as defined within dataset with common morphological labels.**
(CSV)

**S1 Fig. Cell-centered "Knowledge Graph".** We visualized all the pieces of information we had about each cell in a graph. Labels (dark and light blue), modality (e.g. morphological, electrical or genetic classification, dark red), morphological features (green), electrophysiological features (yellow), electrophysiological protocols (gold) were linked to individual cells from either Allen Institute for Brain Science (AIBS) or Blue Brain Project (dataset) for which this information was available.
(PDF)

**S2 Fig. Effect of alpha on datasets overlap (R-value).** Left—R-value computed for different values of $\alpha$. Inset is a zoom of the dashed box. Right—Radius of the global dataset (AIBS +BBP) computed for different values of $\alpha$. Radius computed as the mean of euclidean distance between each of the global dataset instances. Standard deviation from the euclidean distance between each of the instances are displayed in grey.
(PDF)

**S3 Fig. Density profiles of BBP m-types computed from Blue Brain Mouse cell atlas densities combined with probabilistic mapping.** Densities are computed as in Fig 5. Left—Profiles for somatosensory cortex (SSCtx). Right—Profiles for visual primary area (Visp). NGC: Neurogliaform cells; CHC: Chandelier cells; SBC: Small Basket cells; NBC: Nested Basket cells; LBC: Large Basket cells; MC: Martinotti cells; BTC: Bitufted cells; BP: Bipolar cells; DBC: Double Bouquet cells.
(PDF)

**S4 Fig. A**. Mean m-type densities across cortical regions expressed in $mm^{-3}$. The right column shows mean densities for the whole cortex. **B**. Divergence ratio from the mean density over the

whole cortex for each m-type. Ratio was computed as $r_{m-type}^{region} = \frac{mean\ density_{m-type}^{region} - mean\ density_{m-type}^{Isocortex}}{mean\ density_{m-type}^{Isocortex}}$.
(PDF)

**S5 Fig. Dataset cell composition according to different labelling systems.** Cell counts were grouped by Cortical layer, common m-type (as defined in methods), native m-types, native e-types and molecular ID (when available). AIBS on the left and BBP on the right. Dashed lines represent the mean value for each case. The bottom right plot was created by applying the the probabilistic mapping P(marker|m-type) on m-type labels for the BBP dataset. It provides an estimate of the probable marker distribution in the BBP dataset, according to the mapping.
(PDF)

**S6 Fig. E-types mapping and effect on molecular ID. A.** Mapping results when the pipeline was applied using the e-types general description provided by [13] for alpha optimization. AIBS neurons were assigned to one of the four following "common" e-types: Irregular spiking (IR), regular spiking (RS), fast spiking (FS) and adapting (Adapt.). Mapping was done between these "common" e-types and BBP e-types. **B.** Mapping results between molecular ID from AIBS dataset and BBP me-types with e-types labelled as either FS or nFS. Exactly the same morphologies are used to build me-models for both e-types.
(PDF)

**S7 Fig. Principal components from Fig 2 projected on features.** Projections for the e-features space (top), m-features space (middle) and me-features space (bottom) for the not

normalized features, scaled morphologies and Z scaling cases.
(PDF)

## Acknowledgments

We thank Emilie Delattre, Lida Kanari, Alexis Arnaudon, Karin Holm and members of the Molecular Systems team at BBP for helpful comments and suggestions.

## Author Contributions

**Conceptualization:** Yann Roussel, Csaba Verasztó, Dimitri Rodarie.

**Data curation:** Yann Roussel, Csaba Verasztó, Dimitri Rodarie.

**Formal analysis:** Yann Roussel.

**Investigation:** Yann Roussel, Csaba Verasztó, Dimitri Rodarie.

**Methodology:** Yann Roussel, Csaba Verasztó, Dimitri Rodarie.

**Project administration:** Henry Markram, Daniel Keller.

**Software:** Yann Roussel, Tanguy Damart.

**Supervision:** Henry Markram, Daniel Keller.

**Validation:** Yann Roussel, Csaba Verasztó, Dimitri Rodarie, Tanguy Damart, Michael Reimann, Srikanth Ramaswamy, Henry Markram, Daniel Keller.

**Visualization:** Yann Roussel.

**Writing – original draft:** Yann Roussel, Csaba Verasztó, Dimitri Rodarie, Tanguy Damart, Michael Reimann, Srikanth Ramaswamy, Henry Markram, Daniel Keller.

**Writing – review & editing:** Yann Roussel, Csaba Verasztó, Dimitri Rodarie, Tanguy Damart, Michael Reimann, Srikanth Ramaswamy, Henry Markram, Daniel Keller.

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
