## [Decision Letter · Decision Letter 0]

9 Feb 2022

Dear Mr. Roussel,

Thank you very much for submitting your manuscript "Mapping of morpho-electric features to molecular identity of cortical inhibitory neurons" for consideration at PLOS Computational Biology. As with all papers reviewed by the journal, your manuscript was reviewed by members of the editorial board and by several independent reviewers. The reviewers appreciated the attention to an important topic. Based on the reviews, we are likely to accept this manuscript for publication, providing that you modify the manuscript according to the review recommendations.

Sincerely,

Michele Migliore

Associate Editor

PLOS Computational Biology

Daniele Marinazzo

Deputy Editor

PLOS Computational Biology

[LINK]

Reviewer's Responses to Questions

**Comments to the Authors:**

Reviewer #1: 1. Summary of the research and your overall impression

The goal of this work is to develop a model of the cerebral cortex that includes the densities of various types of inhibitory interneurons using datasets that are currently available. The authors used a 3-modal (morphology, ephys, gene expression) dataset of inhibitory neurons from adult mouse visual cortex made by the Allen Institute to organize the morphologically and electrophysiology characterized models of neurons previously produced by the Blue Brain Project from juvenile rat somatosensory cortex. By mapping the morphologically and electrophysiologically defined rat neurons into clusters that matched molecularly-defined mouse neurons, they were able to label the rat neurons with their likely gene expression. Development of a common framework to effectively compare the features of neuron types across species would be useful, not only for the long-term goals of the Blue Brain Project, but also to learn about species differences and similarities. The computational methods are well-described and presented. Further justification for some of the normalization procedures and a more detailed analysis of the biological insights gained from the comparison would extend the work and broaden its impact.

2. Evidence and examples

The introduction states that ‘looking back at such results through the prism of molecular identity would certainly provide new insights.’ However, the manuscript presents few biological insights gained from the development of this computational framework. A discussion of what was learned from this extensive comparison about the similarities and differences between mouse and rat inhibitory neurons would broaden the impact of this work.

The notion that a normalization process can be used to convert a rat neuron into a mouse neuron or vis versa must be supported with a discussion of previous comparative studies. It is as simple as rat neurons being larger and having different input resistance, but otherwise the same as mouse neurons? Normalizing steps for ‘maximum alignment’ will certainly make the rat neurons ‘look’ like mouse neurons, but more justification is needed for doing this. For example, what are the known differences in specific cell types that have been studied extensively in both species?

Along the same lines, the normalizing steps also corrections for differences in cortical areas (V1 vs S1) and development (juvenile vs adult). More support for how this is possible is necessary.

Were the datasets recorded with essentially the same solutions and temperatures? These could also cause significant variability between datasets and should be mentioned.

Additional recordings of rat neurons with single-cell genetics could validate and enrich the study by showing that the predicted molecular identity using the approach described here does indeed match the true molecular identity.

What does alpha or the cross-dataset clustering suggest about the relative variability in morphology and electrophysiology within a molecularly defined cell type? This may be difficult to answer due to the different sizes of the datasets, but a discussion of the variability within cell types in rats and in mice may be useful.

Further analysis of the ‘species-specific’ clusters could reveal new information about whether cell types differ across species. How different are the properties of the cells in the rat-specific cluster from the most similar clusters? By looking at their morphological or electrophysiological features, do they appear unique to rats? If not, an exploration of why they were not matched to any cluster is warranted.

Fig2Aii and 2Aiii appear to be identical.

Reviewer #2: The study by Roussel et al. explores how well multimodal data sets from different species, cortical regions, and ages can be integrated to facilitate the transfer of models created in one system to another. The comparison and linking of cell types across species is an important issue—results of these investigations can help improve cell type definitions, identify common circuits across organisms, and highlight biologically relevant cross-species differences. Here, Roussel et al. intend to populate a model of the mouse isocortex with individual neuron models based on experimental data collected from juvenile rat somatosensory cortex. To accomplish this, they connect a mouse cell atlas (where cells have been grouped into major categories; for interneurons these include Pvalb, Sst, Vip, and others) to the Blue Brain Project (BBP) model database using another data set—a set of morphological and electrophysiological data from the Allen Institute for Brain Science (AIBS) which have molecular identities inferred using transgenic labels and me-type classification. The authors build a unified clustering of both datasets after data normalization/pre-processing and then transfer the molecular labels from AIBS cells to BBP models found in the same cluster. The overall approach taken by the authors is reasonable and appears to be executed well. However, I do have several questions about specific steps taken, as well as some additional questions about how the authors interpret their results.

The authors note that there are several differences between the BBP model data set and the AIBS neuron data set that need to be reconciled. They analyze both sets of data using the same feature analysis tools, but due to differences in stimulus protocols, not all electrophysiological features (e-features) are present for all cells. If I understand correctly, the authors say that they collect the whole set of e-features by finding a core set of e-features for each stimulus amplitude, normalizing the stimulus amplitudes to rheobase, and then seeing which combinations of e-features and stimulus amplitudes are common to the data sets. The authors list 26 e-features in Appendix S1 and say that combining those features with stimulus amplitudes leads to 722 possible e-features (line 258). Using the knowledge graph, they identify e-features that both data sets have in common. But, according to Appendix S3, this turns out to be a list of 26 e-features that are exactly the same as those listed in Appendix S1, without any indication of which stimulus amplitudes are used. Does this mean that only rheobase sweeps were used (even still, they should be labeled as such)? It seems very odd that exactly 26 e-features would be selected by the knowledge graph analysis, and odder still that they have no indication of stimulus amplitude after all the discussion in the Methods about that approach.

Next, the features are pre-processed to reduce the differences between the data sets before clustering both together. It seems like several important questions are not fully addressed in this section. Independently z-scoring each data set and then combining those results implicitly assumes that the mean values of features should be equivalent. However, there are many reasons why the distributions of the features could differ across the data sets — the sampling of cell types could be different, differences in the solutions or temperatures used could affect the properties of cell types in different ways, etc. How do the data sets differ in the first place, and how is z-scoring reconciling those differences? For example, Figure 2A is striking in that nearly all the variation in PC_1 is within AIBS cells and nearly all the variation in PC_2 is among the BBP models — what are the underlying e-features that are driving those differences? Similar, with the m-features, one of the most noticeable things in the non-normalized plot is that there appear to be a few extreme outliers among the BBP models driving the scale of the PCs, which go away after the layer-thickness normalization. How is the normalization changing those properties so dramatically?

The adjustment of the relative weightings (alpha) between e-features and m-features overall made sense, although it could be noted that the me-clustering of Gouwens et al. (2019 and 2020) also used variable weights between e- and m-features, although in those cases the results were combined by consensus clustering rather than optimizing for a single weight. I was not sure I followed the authors' interpretations of all the related results, though (relating to Fig. 3). The authors state that finding consistent values of alpha for the AIBS data set but not the BBP data set is surprising; they would expect higher alpha values for m-only labels. They speculate that it may be due to the smaller size of the AIBS data set and/or an inability to predict the AIBS labels from the chosen features. However, could another factor be that the AIBS e-, m-, and me-labels are overall more congruent with each other than the BBP sets of labels? If, hypothetically, all three sets of labels were identical, one would also expect to find the same alpha value in all cases. More generally, how much does similarity or differences among the label sets affect the range of alphas that could be found? It also doesn't seem obvious that the AIBS label prediction is necessarily worse, since the best cluster homogeneity integral values appear to be comparable between the two data sets.

I am also not certain I fully follow the presentation of the validation results (Table 1). It seems clear that the within-dataset predictions work fairly well, but I'm not sure I understand what to conclude about the cross-data set results. It isn't really explained why one direction (train on BBP, predict AIBS) works substantially better than the other direction. The difference in numbers could of course be at play, but it also seems like predicting BBP m-labels from the joint clusters would be inherently difficult given how split across clusters the BBP m-types are in Fig. 4G. But the within-type prediction of BBP m-types seems to work well (although maybe the clusters are quite different in that case?), as does the prediction of AIBS m-types from the BBP m-type training. In addition, the authors seem surprised that a low alpha value is again found in the AIBS training condition. But the "common m-types" for the AIBS data set are actually based on groups of me-types (according to Table S1), so it seems like there is electrophysiological information available to be used, which might be driving the alpha lower. Why were me-types used for the common labels instead of the AIBS m-types? (As a side note, it also surprised me that the non-Martinotti Sst me-types of ME_Inh_22, 23, and 26 were grouped together with the VIP me-types in this common label system; they seem quite distinct and unrelated).

Overall, I think the authors are tackling an interesting issue of cross-dataset reconciliation. However, I think the study would benefit from more careful investigation of how the inconsistencies manifest — what features are different, how does normalization and z-scoring affect those features, and where are the differences in the validation results arising from? I also feel as though the authors should be clearer that they are linking a set of partially randomized, simulated model data to a set of experimental recordings; for example, the authors speculate that some of the unmatched clusters might represent rat-specific clusters, but perhaps they could also be "modeling-specific" clusters—combinations of e-features and models that are not found in the actual neuron population. Finally, it is a bit anticlimactic that the whole (quite interesting) pipeline is used to assign just the major four interneuron classes to the model cells. There is more interesting molecular diversity present in the mouse cortex than just these basic interneuron divisions (as the authors note and cite relevant papers). I would be more excited to see this pipeline used to connect their large model database to an atlas of more finely defined transcriptomic types (perhaps based on merFISH or similar results) using something like a Patch-seq data set as the linker.

Minor/specific points:

- The section starting on line 323 cites multiple parts of Fig 1D (i through iv) which don't exist. Is it intended to refer to Fig. 3A?

- Fig 2 and the related text could be clearer that the "normalization" refers to the layer-based morphological normalization, since z-scoring is also a form of normalization

- It appears that Figs 4D and 4E are not discussed in the text

- At several points, the text states that excitatory neuron classes are equivalent to layer position; this oversimplifies the situation a bit too much, as it ignores projection type as the other major aspect of traditional excitatory neuron classification (e.g., pyramidal tract-projecting, intratelecephalic projecting, corticothalamic projecting)

- It is stated a few times that the authors use BBP m-types in a figure or discussion instead of the BBP me-types "for clarity" - could they specify why it is clearer? Is it just because there are fewer m-types than me-types, or are there other reasons?

- The feature appendices (S1, S2, S3) also should have brief descriptions of the features so readers do not need to reference to separate documentation to decode the abbreviations

Reviewer #3: See attachment.

Reviewer #4: In this study by Roussel et al., the authors use machine learning algorithms to synthesize morphological, electrophysiological, and genetic marker information to develop integrative expression maps of the interneurons in the cortex. They used the data available from the Blue Brain Project and Allen Brain Atlas to populate their models. To account for the species difference, they limited their data to known genetic markers in rats, and supplemented that information with the richer mouse data. In sum, they demonstrate a feasible approach for integrating morphology, electrophysiology, and molecular identify. Long term goals include adapting the algorithm with internally consistent data sets, and better weighting of individual components.

I do not have any major critiques. The authors acknowledge the practical limitation of the presented data (i.e., different cortical sites, different species). Since the goal of the manuscript was to demonstrate the utility of this type of model, I do not have a problem with data output.

I will note the authors went to great lengths to clearly explain their model construction. I greatly appreciated this effort.

**Have the authors made all data and (if applicable) computational code underlying the findings in their manuscript fully available?**

Reviewer #1: **No: **The link at the end of the results section doesn't work: http://cell-atlas.staging.nexus.ocp.bbp.epfl.ch/

Reviewer #2: Yes

Reviewer #3: Yes

Reviewer #4: Yes

PLOS authors have the option to publish the peer review history of their article (what does this mean?). If published, this will include your full peer review and any attached files.

Reviewer #1: No

Reviewer #2: No

Reviewer #3: **Yes: **Niccolo' Calcini

Reviewer #4: No

Figure Files:

Data Requirements:

Reproducibility:

References:

---

## [Editor Report · Decision Letter 1]

26 Mar 2022

Dear Mr. Roussel,

We are pleased to inform you that your manuscript 'Mapping of morpho-electric features to molecular identity of cortical inhibitory neurons' has been provisionally accepted for publication in PLOS Computational Biology.

Best regards,

Michele Migliore

Associate Editor

PLOS Computational Biology

Daniele Marinazzo

Deputy Editor

PLOS Computational Biology

---

## [Editor Report · Acceptance letter]

13 Dec 2022

PCOMPBIOL-D-22-00099R1 

Mapping of morpho-electric features to molecular identity of cortical inhibitory neurons

Dear Dr Roussel,

I am pleased to inform you that your manuscript has been formally accepted for publication in PLOS Computational Biology. Your manuscript is now with our production department and you will be notified of the publication date in due course.

With kind regards,

Zsofi Zombor
